# Different Effects of Albumin and Hydroxyethyl Starch on Low Molecular-Weight Solute Permeation through Sodium Hyaluronic Acid Solution

**DOI:** 10.3390/polym13040514

**Published:** 2021-02-09

**Authors:** Tsuneo Tatara

**Affiliations:** Department of Anesthesiology and Pain Medicine, Hyogo College of Medicine, 1-1 Mukogawa-cho, Nishinomiya, Hyogo 663-8501, Japan; ttatara@hyo-med.ac.jp

**Keywords:** hyaluronic acid, permeation, hydroxyethyl starch, inflammation

## Abstract

Hyaluronic acid (HA), a high-molecular-weight linear polysaccharide, restricts solute transport through the interstitial space. Albumin and hydroxyethyl starch (HES) solutions are used to correct the decrease of blood volume during surgery, but may leak into the interstitial space under inflammation conditions. Given the possibility that the structure of HA may be affected by adjacent macromolecules, this study tested whether albumin and HES (*M*_w_ 130,000) exert different effects on solute permeation through sodium hyaluronic acid (NaHA: *M*_w_ 1.3 × 10^6^) solution. To this end, permeation of Orange G, a synthetic azo dye (*M*_w_ 452), into NaHA solutions containing albumin or HES over time was assessed. The amount of time it took for the relative absorbance of Orange G to reach 0.3 (*T*_0.3_) was determined in each NaHA solution relative to the reference solution (i.e., colloid solution without NaHA). Relative *T*_0.3_ values of albumin were larger than those of HES for 0.1% NaHA solution (3.33 ± 0.69 vs. 1.16 ± 0.08, *p* = 0.006, *n* = 3) and 0.2% NaHA solution (1.95 ± 0.32 vs. 0.92 ± 0.27, *p* = 0.013, *n* = 3). This finding may help in the selection of an appropriate colloid solution to control drug delivery into the interstitial space of cancer tissue under inflammation conditions.

## 1. Introduction

Hyaluronic acid (HA), a glycosaminoglycan found in the interstitial space [1,2], is a high-molecular-weight linear polysaccharide polymer consisting of repeating disaccharide units of *β*-d-glucuronic acid and *N*-acetyl-*β*-d-glucosamine [3]. HA plays a major role in many biological processes including cell signaling, wound healing, and matrix organization [4]. HA forms a fibrous gel-like network at semidiluted concentrations (approximately 0.6 mg/mL) [5]. Since the concentration of HA in the interstitial space is in the range of 1–10 mg/mL, HA is presumed to exist in a physically entangled state in the interstitial space [1,2], which restricts solute transport [6,7]. This is a clinically important aspect, as diffusion properties of solutes in the interstitial space influence the inflammatory process [8] and drug delivery in diseases such as cancer [9].

In cancer patients undergoing major surgery, adequate fluid administration using colloid solutions is essential for correcting the decrease of blood volume due to ongoing hemorrhage and inflammation during surgery. Albumin solution, which is prepared from pooled human plasma, is suitable for this purpose, but its limited availability and high cost are major issues. Accordingly, hydroxyethyl starch (HES) solution, a synthetic colloid solution modified from waxy maize starch largely composed of highly branched amylopectin, is now widely used as an alternative to albumin solution [10]. Nevertheless, as inflammation caused by surgical injury increases capillary permeability to macromolecules, intravenously infused colloids may leak into the interstitial space [11]. It is thus necessary to investigate whether leakage of albumin and HES into the interstitial space could modify the structure of HA and thereby affect interstitial solute transport.

Albumin stabilizes HA structure by binding to HA via weak electrostatic forces and hydrogen bonding [12,13]. In a recent study, HES, but not albumin, was found to decrease the intrinsic viscosity of sodium hyaluronic acid (NaHA), suggesting that HES may locally restrict NaHA diffusion via hydrogen bonding with NaHA chains [14]. However, as the intrinsic viscosity of NaHA is obtained by extrapolating reduced viscosity of NaHA solution to zero concentration of NaHA, the effects of albumin and HES on the physical properties of semidiluted NaHA solution, in which the interaction of NaHA chains among themselves cannot be neglected, must be investigated.

On this basis, the present study hypothesized that albumin and HES exert different effects on NaHA diffusion and thus on the permeation of small solutes through NaHA solution at physiological concentration. Since HES causes the formation of clusters of NaHA chain segments by restricting the spreading of NaHA chains, the resultant decrease of friction force between NaHA chains and the permeating solute might accelerate solute permeation into NaHA solution compared to albumin [15]. To test this, the present study examined how albumin and HES affect small solute permeation into NaHA solution. Moreover, to characterize the diffusion properties of NaHA, the osmotic swelling pressure, hydraulic permeability coefficient, and dynamic shear moduli of NaHA solution in the presence of albumin and HES were measured. The finding that albumin restricted low molecular-weight solute permeation through NaHA solution to a greater degree than HES may help in the selection of an appropriate colloid solution to control drug delivery into the interstitial space of cancer tissue under inflammation conditions.

## 2. Materials and Methods 

### 2.1. Materials

This study used a bovine serum albumin solution and a commercially available 6% (*w*/*v*) HES solution of weight-average *M*_w_ 130,000 (waxy maize starch-based HES: molar substitution, 0.41; C_2_/C_6_ ratio, 9:1; Voluven^®^, Fresenius Kabi, Bad Homburg, Germany) [10]. These colloid solutions were diluted to desired concentrations with phosphate buffered saline (PBS) (pH 7.4).

NaHA from *Streptococcus equil* (*M*_w_ 1.3 × 10^6^) [14] and bovine serum albumin (*M*_w_ 66,000) were purchased from Sigma-Aldrich (St. Louis, MO, USA). Orange G, a synthetic azo dye (*M*_w_ 452), was purchased from FUJIFILM Wako Pure Chemical (Osaka, Japan). Chemical structures of NaHA, albumin, HES, and Orange G are shown in Appendix A. All reactions were carried out using purified water from a Millipore Milli-Q purification system (Merck Millipore, Burlington, MA, USA). Colloid solutions were filtered through a 0.45-µm membrane prior to analysis.

### 2.2. Permeation of Orange G into NaHA Solutions

NaHA was dissolved in albumin or HES solution and left for 24 h to solidify with gentle stirring at room temperature (25 °C). NaHA solutions were prepared at the following concentrations: (i) 0.1% (*w*/*w*) NaHA in 1.1% (*w*/*v*) albumin or HES solution; (ii) 0.2% (*w*/*w*) NaHA in 1% (*w*/*v*) albumin or HES solution; and (iii) 0.4% (*w*/*w*) NaHA in 0.8% (*w*/*v*) albumin or HES solution. In all solutions, the total concentration of solutes (i.e., NaHA plus colloids) was kept constant so as to minimize the effects of specific gravity of the solution on solute permeation. Colloid solutions without NaHA (1.2% (*w*/*v*) albumin/HES) were used as references. These NaHA concentrations fall within the range observed in the interstitial space (i.e., 0.1–1%) [1,2]. A HES concentration of 1% was chosen because this concentration falls within the range of plasma concentration in clinical settings (i.e., 1–2%) [10].

PBS solution containing 0.1 mL 0.01% (*w*/*v*) Orange G was gently poured over 1.4 mL NaHA solution in a semimicro ultraviolet (UV) cuvette (12.5 mm square; inner width, 4 mm; height, 45 mm; path length, 10 mm) (Figure 1). The amount of Orange G that permeated into NaHA solution was determined every 6 min for 20 h by measuring absorbance at 478 nm using a UV spectrometer (Model UV-1850; Shimadzu, Kyoto, Japan) (Figure 1). The temperature of the UV cuvette was maintained at 37 °C with a temperature control unit (Model TCC-100; Shimadzu, Kyoto, Japan). The top of the UV cuvette was sealed with laboratory film (Parafilm^®^, Bemis Flexible packaging, Chicago, IL, USA) to prevent sample evaporation. Experiments were carried out at least in triplicate. Results were expressed as relative absorbance (i.e., relative to the absorbance value calculated assuming that Orange G is uniformly distributed into NaHA solution in the cuvette).

For the quantitative analysis of Orange G permeation into NaHA solution, the amount of time it took for the relative absorbance of Orange G to reach 0.3 (*T*_0.__3_ in h) was determined in each NaHA solution relative to the reference solution (i.e., colloid solution without NaHA). Results were compared between NaHA solutions containing albumin and those containing HES with the unpaired *t*-test.

### 2.3. Effects of Albumin and HES on Osmotic Swelling Pressure of NaHA Solution

NaHA was dissolved in PBS to prepare a 1% (*w*/*w*) solution and left for 24 h to solidify with gentle stirring at room temperature. The osmotic swelling pressure of 1% NaHA solution was measured using an osmotic flow cell as previously reported [16,17]. The basic principle of the osmotic flow cell is as follows: two fluid chambers are separated by a semipermeable membrane (molecular weight cut-off, 300,000; ultrafiltration membrane disk PBMK; Merck Millipore, Burlington, MA, USA) (Figure 2), with one of the chambers serving as the sample chamber (filled with 0.5 mL 1% NaHA in PBS) and the other as the reference chamber (filled with PBS, 2% (*w*/*v*) albumin, or 2% (*w*/*v*) HES), which is connected via a narrow channel (0.5 mm diameter) to a manometric chamber fitted with an electronic pressure transducer (PGM-02KG, Kyowa, Tokyo, Japan). The volume of colloid-diffusing fluid space (i.e., reference and manometric chambers, and the channel connecting them) is 1.5 mL. Fluid movement from the reference chamber toward the sample chamber due to the osmotic swelling force of NaHA solution creates a negative hydrostatic pressure in the reference chamber, which equals the osmotic swelling pressure of NaHA solution against test colloid solutions. Hydrostatic pressure in the reference chamber was recorded every 10 s for 20 h using LabVIEW^®^ (National Instruments, Austin, TX, USA). Experiments were repeated four times at room temperature.

Values of hydrostatic pressure in the reference chamber at 2 h, 5 h, 10 h, 15 h, and 20 h were compared between PBS, albumin, and HES by one-way analysis of variance (ANOVA).

### 2.4. Effects of Albumin and HES on Darcy’s Permeability Coefficient of NaHA Solution

NaHA was dissolved in PBS to prepare 0.1% (*w*/*w*), 0.3% (*w*/*w*), and 0.5% (*w*/*w*) solutions and left for 24 h to solidify with gentle stirring at room temperature. Darcy’s permeability coefficient (*K*) of each NaHA solution was measured using the ultra-fast double-sided reusable sample dialyzer^TM^ (outer diameter 2.5 cm, chamber volume 500 μL: 7404-5001D, Harvard Apparatus, Holliston, MA, USA) (Figure 3). The system interposed 0.5 mL of NaHA solution between two precut cellulose acetate dialysis membranes (molecular weight cut-off 300 kDa: 7403-CA300K, Harvard Apparatus, Holliston, MA, USA). PBS or 2% (*w*/*v*) colloid (i.e., albumin, HES) solution was infused into the dialysis chamber containing NaHA solution via a connecting tube with an automatic infusion pump (FP-2200, MELQUEST, Toyama, Japan) at a constant rate (*Q*) ranging from 1 to 20 μL/min (Figure 3).

The pressure tube was connected to a manometric chamber fitted with an electronic pressure transducer (PGM-1KG, Kyowa, Tokyo, Japan). Hydrostatic pressure in the pressure tube was continuously recorded every 50 s using LabVIEW^®^ (National Instruments, Austin, TX, USA), which was found to steeply increase with time and then reach a plateau over 1–40 h, depending on NaHA concentration.

The value of *K* (cm^2^) was calculated according to Darcy’s law, as follows [2]:(1)$$K=\frac{l}{A}\cdot \frac{\eta }{\left(\Delta P/\Delta Q\right)},$$
where *A* is the membrane surface area available for fluid filtration (0.75^2^ × 3.14 = 1.77 cm^2^), *l* is the thickness of NaHA solution in the dialysis chamber (0.28 cm, calculated as 0.5/*A*), *η* is the viscosity of infused fluid, and *P* is the hydrostatic pressure in the pressure tube at plateau. Values of Δ*P*/Δ*Q* were obtained by the linear fitting procedure for *P* values plotted against *Q* values using GraphPad Prism 5^®^ software (GraphPad Software Inc., San Diego, CA, USA). Values of *η* at 25 °C (mPa·s) for PBS, 2% albumin, and 2% HES were 0.95, 0.99, and 1.39, respectively, which were measured using a vibrational viscometer (Model SV-1A; A&D, Inc., Tokyo, Japan) equipped with a temperature controller (Model NCB-1200; EYELA, Tokyo, Japan) kept at 25 °C.

### 2.5. Effects of Albumin and HES on Dynamic Shear Moduli of NaHA Solution

NaHA was dissolved in PBS to prepare 0.5% (*w*/*w*) solution with or without 3% (*w*/*v*) and 6% (*w*/*v*) albumin or HES, and left for 24 h to solidify with gentle stirring at room temperature. Dynamic shear moduli of each NaHA solution were measured by small amplitude oscillatory shear experiments over the frequency range of 0.1–10 Hz with a rotational rheometer (HAAKE Viscotester iQ Air, Thermo Fisher Scientific, Waltham, MA, USA) equipped with a Peltier temperature control module kept at 37 °C. The software HAAKE RheoWin (ver. 4.86, Thermo Fisher Scientific, Waltham, MA, USA) was used to determine storage (*G*′) and loss (*G*″) shear moduli of NaHA solution at each frequency. Cone geometry with a diameter of 6 cm and core angle of 2° was used. Prior to frequency sweep experiments, strain amplitude was confirmed by strain sweep experiments to be sufficiently small to provide a linear material response at all investigated frequencies. A solvent trap was used to avoid evaporation of samples during experiments. Experiments were carried out at least six times.

### 2.6. Statistical Analysis

Statistical analyses were performed using SigmaPlot 13 (Systat Software Inc., Chicago, IL, USA) software. *p* < 0.05 was considered statistically significant.

## 3. Results

### 3.1. Permeation of Orange G into NaHA Solution

Direct visualization of NaHA solution in the UV cuvette showed that Orange G permeated into NaHA in HES solution rapidly compared to NaHA in albumin solution (Appendix A). The presence of albumin significantly decreased permeation of Orange G into NaHA solution almost by half compared to the reference, regardless of NaHA concentration (Figure 4a). On the other hand, the presence of HES decreased permeation of Orange G into NaHA solution at 0.4%, but not at 0.1% or 0.2%, compared to the reference (Figure 4b).

Relative *T*_0.3_ values of albumin were significantly larger than those of HES for the 0.1% and 0.2% NaHA solutions, but did not significantly differ between albumin and HES for the 0.4% NaHA solution (Table 1).

### 3.2. Effects of Albumin and HES on Osmotic Swelling Pressure of NaHA Solution

Hydrostatic pressure of PBS, 2% albumin, and 2% HES solutions measured in the reference chamber steeply decreased within 30 min, and thereafter gradually decreased until equilibrium was reached at roughly 15 h (Figure 5). HES had a higher hydrostatic pressure in the reference chamber at 2 h compared to PBS (*p* = 0.022) and albumin (*p* = 0.021) and at 5 h compared to PBS (*p* = 0.029). No significant difference in hydrostatic pressure was noted between PBS, albumin, and HES at other time points.

### 3.3. Effects of Albumin and HES on Darcy’s Permeability Coefficient of NaHA Solution

Fluid infusion into NaHA solution increased the hydrostatic pressure in the pressure tube over time, reaching a plateau (Figure 6a). It was thus confirmed that the flow–pressure relationship obeyed Darcy’s law. *P* values at plateau were plotted against *Q* values to obtain Δ*P*/Δ*Q* values using a linear fitting procedure, which were then used to determine *K* values of NaHA solutions (Figure 6b).

*K* values decreased as the concentration of NaHA increased for both PBS and colloid solutions. *K* values of colloid solutions were lower than those of PBS solution at all concentrations of NaHA. Compared to albumin, higher *K* values were observed for HES at 0.1% NaHA, while values were lower at 0.3% and 0.5% NaHA (Figure 7). As a result, the slope of *K* values against NaHA concentrations was steeper for HES than that for albumin.

### 3.4. Effects of Albumin and HES on Dynamic Shear Moduli of NaHA Solution

While the presence of 3% albumin in NaHA solution significantly increased *G*′ and *G*″ of NaHA solution relative to PBS at low frequency (i.e., <1 Hz), *G*′ and *G*″ of NaHA solution were similar for PBS and 3% HES (Figure 8a). Nevertheless, the increase in HES concentration from 3% to 6% significantly increased *G*′ and *G*″ of NaHA solution, resulting in larger *G*′ and *G*″ of NaHA solution compared to PBS and 6% albumin at all frequencies examined (Figure 8b).

## 4. Discussion

### 4.1. Permeation of Orange G into NaHA Solution

This study demonstrated that albumin and HES exerted different effects on the permeation of Orange G into NaHA solution. The presence of albumin significantly decreased Orange G permeation at ≥0.1% NaHA (Figure 4a), while the presence of HES did not significantly affect Orange G permeation into NaHA solution at concentrations up to 0.2% (Figure 4b). According to Ogston’s theory on the distribution of spaces in a uniform random suspension of fibers [18], the pore radius of NaHA solution (*r*_p_) is expressed as follows [19]:(2)$$r_{p}=\left(r_{f}/4\right)\sqrt{\pi /\varphi_{f}},$$
where *r*_f_ and *φ*_f_ are the radius and volume fraction of NaHA, respectively, and
(3)$$\varphi_{f}=c\cdot \nu ,$$
where *c* and *ν* are the concentration (g/cm^3^) and partial specific volume of NaHA (0.653 cm^3^/g in 0.2 M NaCl solution [2]), respectively. The reported *r*_f_ value of NaHA is 0.55 nm [2]. Therefore, the pore radii of 0.1%, 0.2%, and 0.4% NaHA solutions according to Equations (2) and (3) are 9.5 nm, 6.7 nm, and 4.8 nm, respectively. Given that the Stokes–Einstein radii of albumin and HES molecules are reported to be 3.5 nm [20] and 6.1 nm [21], respectively, albumin can permeate through the pores of NaHA solution at all three concentrations, whereas HES molecules can permeate through the pores of NaHA solution at 0.1% and 0.2%, but not 0.4%.

The partitioning of albumin into NaHA pores explains its obstructive effect, and hence, the restriction of Orange G permeation into NaHA solution [22]. As for HES, despite the assumption that HES can partition into the hydrodynamic pores of NaHA solution at concentrations of 0.1% and 0.2%, no restriction of Orange G permeation into NaHA solution was observed. One possible explanation for this discrepancy is the difference between the global structure of albumin and HES. First, the spherical structure of albumin may retard Orange G diffusion compared to the linear polymer structure of HES by increasing frictional resistance between Orange G and albumin. Second, the electrostatic interaction of positively charged amino acid residues of albumin with negatively charged Orange G could slow Orange G diffusion compared to electrically neutral HES (Appendix A). Finally, the inhomogeneous structure of NaHA solution may have contributed at least in part to the observed discrepancy. The uneven distribution of NaHA chain segments creates regions of low NaHA concentration and high NaHA concentration (i.e., clusters) due to the entanglement of NaHA chain segments [15]. Given that HES accelerates this uneven NaHA chain segment distribution by restricting NaHA diffusion, the resultant decrease of friction force between NaHA chains and permeating solute [15] might have cancelled out the obstructive effect of HES.

### 4.2. Effects of Albumin and HES on Osmotic Swelling Pressure and Darcy’s Permeability Coefficient of NaHA Solution

The mean osmotic pressure (i.e., decrease of hydrostatic pressure in the reference chamber at 20 h from baseline) of 1% NaHA in PBS was 0.69 kPa, which is comparable to that of 1% HA (*M*_w_ 1.5 × 10^6^) in PBS (pH 7.6) reported previously (0.69 kPa) [23], demonstrating the acceptable accuracy of the measuring system. The significantly smaller osmotic swelling pressure (i.e., decrease of hydrostatic pressure in the reference chamber from baseline) of 1% NaHA against HES at 2 h compared to albumin suggests that HES slows osmotic swelling of NaHA arising from the collective diffusion of NaHA (Figure 5) [15].

According to the hydrodynamic modeling of HA polymer network, *K* (Darcy’s permeability coefficient, cm^2^) is expressed as follows [24]:(4)$$K=5.4\times 10^{-16}\cdot \frac{\ln(1/c)-0.18}{c},$$
where *c* is the concentration (g/cm^3^) of NaHA. Consistent with a previous study, which measured flow rates at given pressure drops (i.e., up to 10 kPa) across HA solutions of 0.05–1.5% (*w*/*w*) [24], the *K* values obtained in the present study were comparable with those calculated by Equation (4) for 0.3% NaHA in PBS (8.1 cm^2^ vs. 10.1 cm^2^) and 0.5% NaHA in PBS (4.4 cm^2^ vs. 5.5 cm^2^) (Figure 7). The *K* value for 0.1% NaHA in PBS, which was two-fold smaller than that calculated by Equation (4) (17 cm^2^ vs. 36 cm^2^), may not be unreasonable given that the contribution of dialysis membranes to the overall *K* value is not negligible compared to the *K* value for 0.1% NaHA in PBS. The steep slope of *K* values against NaHA concentrations for HES compared to albumin (Figure 7) suggests that frictional resistance between HES and NaHA chains increased with increasing NaHA concentration to a larger extent compared to albumin.

Collective diffusion is determined by the balance between the thermodynamic force, which drives spreading of solute particles and the frictional force, which resists the spreading [15]:(5)$$D_{c}\left(n\right)=\frac{n}{\xi }\cdot \frac{\partial \Pi }{\partial n},$$
where *D_c_*(*n*) is the collective diffusion constant of solute, *n* is the particle density, *ξ* is the friction coefficient, and ∂*Π*/∂*n* is the osmotic pressure. Given that HES increases frictional resistance, as reflected by the steep slope of *K* values against NaHA concentrations (i.e., larger *ξ*), HES might have restricted the collective diffusion of NaHA as shown by slowed osmotic swelling of NaHA by HES aforementioned, thereby accelerating the uneven distribution of NaHA chain segments.

### 4.3. Effects of Albumin and HES on Dynamic Shear Moduli of NaHA Solution

*G*′ and *G*″ of NaHA solution were significantly increased in the presence of 6% HES compared to PBS or 6% albumin (Figure 8b). HA chain segments are considered to exist in a continuous equilibrium between stiff and flexible states, and their local conformational ordering is maintained by continuous forming and breaking of hydrogen bonds [25]. In particular, hydrogen bonds between adjacent saccharides largely contribute to the intrinsic stiffness of HA under physiological electrolyte concentrations and pH conditions [26].

HES-induced increases in *G*′ and *G*″ of NaHA solution were consistent with sugar-induced increases in *G*′ and *G*″ of HA previously reported, and likely resulted from the increase of bond angle restriction and the creation of new hydrogen bonds [25]. As *G*′ and *G*″ of NaHA solution at frequency <100 Hz reflect the orientational dynamics of NaHA chains [27], 6% HES might increase the number of elastically active NaHA chains to a greater extent than 6% albumin by strengthening the transient NaHA network. HES at 6% exceeds the plasma HES concentration used in clinical settings (1–2%). Nevertheless, it is possible that the viscoelastic property of NaHA at lower concentrations (e.g., 0.2%) is readily affected by lower concentrations of HES (e.g., 2%), although the rotational rheometer used in the present study could not examine dynamic shear moduli of NaHA solution of lower concentrations due to minimum torque constraints.

### 4.4. Implications

The present study demonstrated that albumin restricted Orange G permeation through NaHA solution to a greater degree than HES. The different effects of albumin and HES may be related to the uneven NaHA chain segment distribution arising from the restriction of NaHA diffusion by HES. Since the molecular weights of antibiotics, anti-inflammatory drugs, and anticancer drugs range from hundreds to thousands [28,29,30], i.e., comparable with that of Orange G, this finding may help in selecting an optimal colloid solution for effective drug delivery in the treatment of cancer patients undergoing major surgery. If the treatment requires rapid onset of drugs, HES solution may allow for their rapid distribution in the interstitial space. In contrast, given that anticancer drugs must remain in the interstitial space for an extended period, albumin solution may be preferable because it may delay the transport of the drugs from the interstitial space to lymphatic vessels by restricting diffusion through the interstitial space. As interstitial resistance to solute permeation is influenced not only by finer fibrous molecules such as HA, but also coarse fixed elements such as collagen fibrils [2], these scenarios warrant further investigation in in vivo studies. Studies on the contribution of hydrogen bonding of colloids with NaHA chains to solute permeation properties are also warranted.

## Figures and Tables

**Figure 1 polymers-13-00514-f001:**
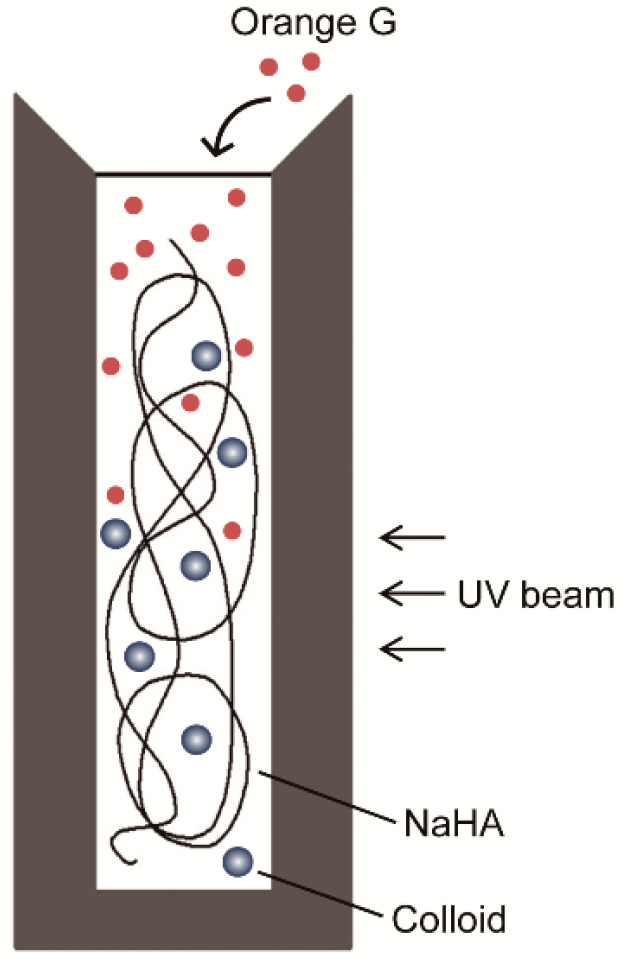
Experimental set-up for the measurement of solute permeation into sodium hyaluronic acid (NaHA) solution. A solution containing Orange G (0.1 mL) was poured over NaHA solution containing colloid (i.e., albumin or hydroxyethyl starch) in an ultraviolet (UV) cuvette.

**Figure 2 polymers-13-00514-f002:**
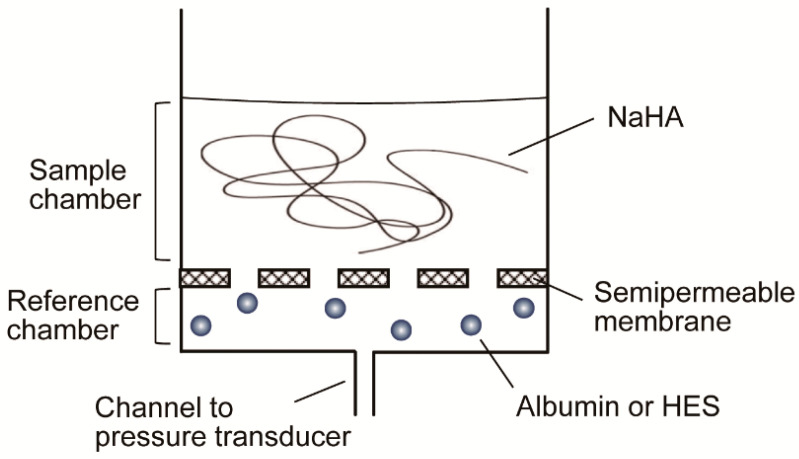
Schematic of an osmotic flow cell. Fluid flows from the reference chamber to the sample chamber due to osmotic swelling of sodium hyaluronic acid (NaHA) in phosphate buffered saline (PBS). The sample chamber was filled with 0.5 mL 1% NaHA solution, and the reference chamber was filled with PBS, 2% albumin, or 2% hydroxyethyl starch solution (HES). A semipermeable membrane with a molecular weight cut-off of 300,000 was used.

**Figure 3 polymers-13-00514-f003:**
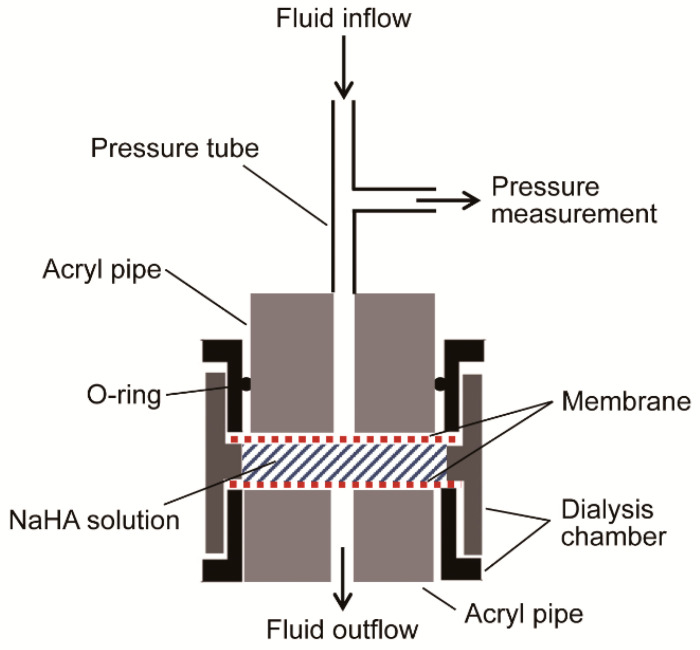
Experimental set-up for the measurement of Darcy’s permeability coefficient of sodium hyaluronic acid (NaHA) solution. The fluid (i.e., phosphate buffered saline or colloid solution) was infused into the dialysis chamber containing NaHA solution at a constant rate, and hydrostatic pressure in the pressure tube was continuously recorded.

**Figure 4 polymers-13-00514-f004:**
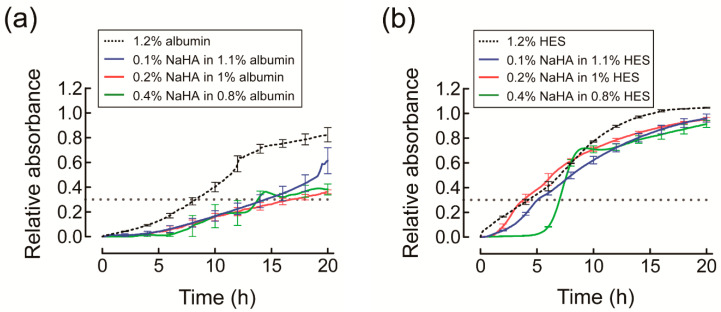
Time course of relative absorbance of Orange G permeating into different concentrations of sodium hyaluronic acid (NaHA) solution containing (**a**) albumin or (**b**) hydroxyethyl starch (HES). Relative absorbance of Orange G is expressed as a value relative to that calculated assuming that Orange G is uniformly distributed into NaHA solution in the ultraviolet cuvette. Experiments were carried out at least in triplicate. For simplicity, values are presented as mean and standard error for every 2 h. The dotted horizontal line indicates a relative absorbance of 0.3.

**Figure 5 polymers-13-00514-f005:**
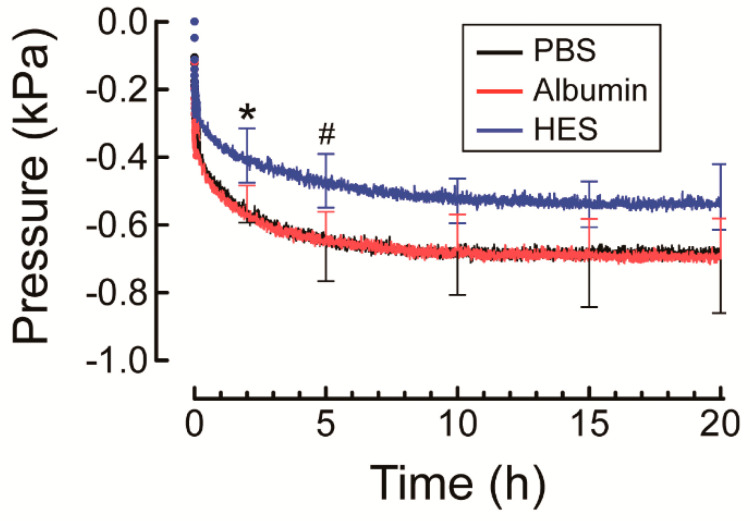
Comparison of changes over time of hydrostatic pressure in the reference chamber of the osmotic flow cell for 1% sodium hyaluronic acid solution (i.e., osmotic swelling pressure) against phosphate buffered saline (PBS), 2% albumin, and 2% hydroxyethyl starch (HES) solution. Experiments were carried out at 25 °C. For simplicity, values are presented as mean and standard deviation for every 5 h (*n* = 4). *: significantly different from PBS and albumin at the same time point; #: significantly different from PBS at the same time point.

**Figure 6 polymers-13-00514-f006:**
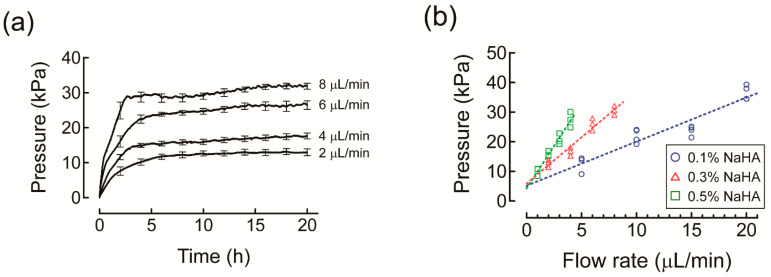
(**a**) Time course of changes in hydrostatic pressure in the pressure tube for 0.3% sodium hyaluronic acid solution (NaHA) at different infusion rates of phosphate buffered saline (PBS) in the measurement of Darcy’s permeability coefficient of NaHA solution. Experiments were carried out at 25 °C. For simplicity, values are presented as mean and standard error for every 2 h. (**b**) Relationship between infusion rates of PBS and hydrostatic pressure in the pressure tube at plateau for different concentrations of NaHA solution fitted with linear regression lines (dashed lines).

**Figure 7 polymers-13-00514-f007:**
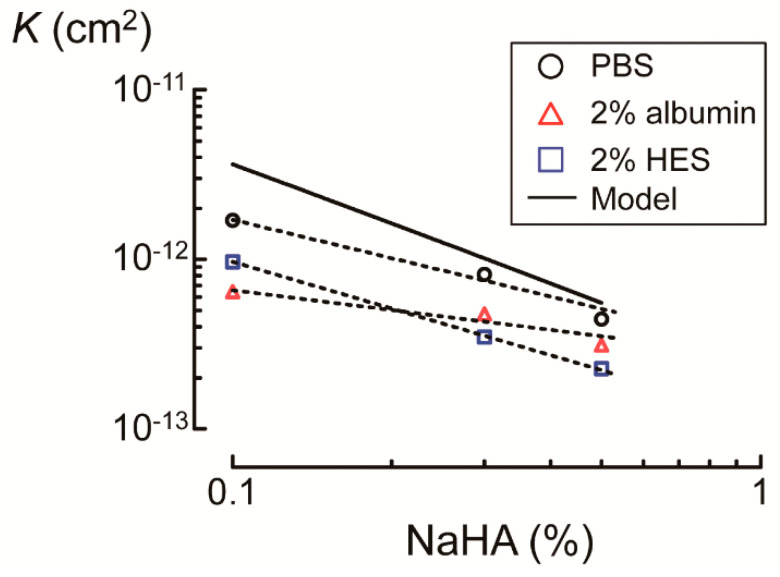
Comparison of Darcy’s permeability coefficients (*K*) of sodium hyaluronic acid (NaHA) solution when phosphate buffered saline (PBS) or 2% colloid solution (i.e., albumin or hydroxyethyl starch (HES)) was infused. Solid line denotes *K* values calculated by Equation (4) according to the hydrodynamic model of hyaluronic acid polymer networks. Dotted lines denote linear regression lines between logarithmic values of NaHA concentrations and *K*.

**Figure 8 polymers-13-00514-f008:**
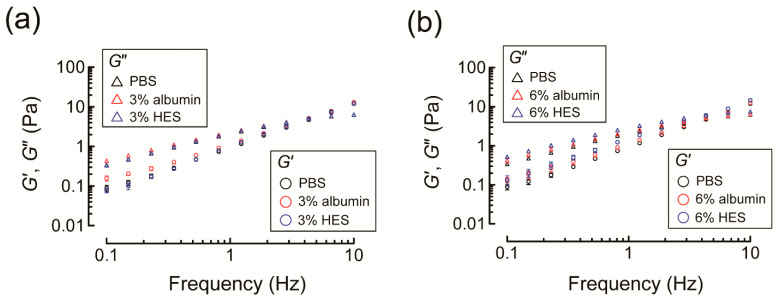
(**a**) Comparison of storage (*G*′) and loss (*G*″) shear moduli of 0.5% sodium hyaluronic acid (NaHA) dissolved in phosphate buffered saline (PBS) or 3% colloid solution (i.e., albumin or hydroxyethyl starch (HES)). (**b**) Comparison of *G*′ and *G*″ of 0.5% NaHA dissolved in PBS or 6% colloid solution (i.e., albumin or HES). Values are expressed as mean and standard deviation (*n* = 6 or 7).

**Table 1 polymers-13-00514-t001:** Relative time ^1^ at which relative Orange G absorbance reached 0.3 as it permeated into sodium hyaluronic acid solution (NaHA) containing albumin or hydroxyethyl starch (HES).

Test Solution	Colloid		*p* Value ^2^
	Albumin	HES	
0.1% NaHA in 1.1% colloid	3.33 ± 0.69	1.16 ± 0.08	0.006
0.2% NaHA in 1% colloid	1.95 ± 0.32	0.92 ± 0.27	0.013
0.4% NaHA in 0.8% colloid	3.19 ± 1.33	1.58 ± 0.03	0.11

^1^ Time relative to reference (1.2% colloid). ^2^ Albumin versus HES. Values are shown as mean ± standard deviation (*n* = 3).

## Data Availability

The data presented in this study are available on request from the corresponding author.

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
