# Peer review of "Different Effects of Albumin and Hydroxyethyl Starch on Low Molecular-Weight Solute Permeation through Sodium Hyaluronic Acid Solution"

_polymers, 2021, doi:10.3390/polym13040514_

Round 1

Reviewer 1 Report

The manuscript investigate the influence of hyaluronic acid (HA) on the restriction of solute transport through interstitial space that is critical for controlling the delivery of active species such as drugs in the space. The manuscript is interesting and can be acceptable after addressing some issues.

  • Chemical structures of albumin, HA and HES should be provided as a Scheme. Currently, it is hard to immediately capture the influence of molecular structures on performances.
  • Intermolecular interactions between chemicals should be discussed based on the structures in the above and in aspect of their assemblies.
  • Direct visualization for Orange G permeation is necessary to prove the homogeneous or inhomogeneous distribution of the dye in the medium. Optical imaging can provide the staining effect of the dye. Currently, the mechanism is too speculated.
  • In Figure 4, the experiments under controlled conditions are necessary. For example, the influence of increasing Albumin or HES concentrations at a fixed NaHA concentration. Currently, both control factors are varying, which make data analysis difficult.
  • Experimental proofs for porous NaHA gel structures are necessary. It is hard to believe the porous structures in the conditions suggested in the manuscript.
  • English editing is necessary.

Author Response

To Reviewer #1:

Thank you for reviewing the manuscript and providing helpful suggestions, which have helped improve the revised manuscript. As described in the point-by-point responses below, all issues raised have been addressed in the revised manuscript.

The manuscript investigate the influence of hyaluronic acid (HA) on the restriction of solute transport through interstitial space that is critical for controlling the delivery of active species such as drugs in the space. The manuscript is interesting and can be acceptable after addressing some issues.

  1. Chemical structures of albumin, HA and HES should be provided as a Scheme. Currently, it is hard to immediately capture the influence of molecular structures on performances.

Response to the comment:

As suggested, the schemes of chemical structures of albumin, hyaluronic acid (HA), hydroxyethyl starch (HES), and Orange G have been added as a Supplemental Figure (Figure S1).

  1. Intermolecular interactions between chemicals should be discussed based on the structures in the above and in aspect of their assemblies.

Response to the comment:

As suggested, brief comments on possible interactions between chemicals have been added to the Discussion section (page 10: line 283-289), as follows:

“One possible explanation for this discrepancy is the difference between the global structure of albumin and HES. First, the spherical structure of albumin may retard Orange G diffusion compared to the linear polymer structure of HES by increasing frictional resistance between Orange G and albumin. Second, the electrostatic interaction of positively-charged amino acid residues of albumin with negatively-charged Orange G could slow Orange G diffusion compared to electrically neutral HES (Figure S1).”

  1. Direct visualization for Orange G permeation is necessary to prove the homogeneous or inhomogeneous distribution of the dye in the medium. Optical imaging can provide the staining effect of the dye. Currently, the mechanism is too speculated.

Response to the comment:

I agree, but unfortunately do not have an experimental apparatus for optical imaging of dye (i.e., Orange G) permeation into sodium hyaluronic acid (NaHA) solution. Orange G macroscopically appeared to distribute homogenously into all NaHA solutions tested. As the precise mechanisms underlying the differences in Orange G permeation into NaHA solutions between albumin and HES cannot be elucidated in this study, the description of the mechanism has been revised (page 10: line 289-290), as follows:

“Finally, the inhomogeneous structure of NaHA solutions may have contributed at least in part to the observed discrepancy.”

  1. In Figure 4, the experiments under controlled conditions are necessary. For example, the influence of increasing Albumin or HES concentrations at a fixed NaHA concentration. Currently, both control factors are varying, which make data analysis difficult.

Response to the comment:

I agree that albumin or HES concentrations should be altered while fixing the NaHA concentration. Nevertheless, in the experimental setup for Orange G permeation into NaHA solutions in the present study, the increase of specific gravity of the medium slows Orange G permeation into NaHA solutions. For example, it was found that Orange G permeation into PBS was slowed compared to permeation into distilled water. On this basis, the total concentration of solutes (i.e., NaHA plus colloids) was forced to be constant so as to minimize the effects of specific gravity of the solution on solute permeation. This is described in the Methods section (page 3: line 88-89). I also believe this issue does not significantly impact the conclusion relating to the different effects of albumin and HES on dye permeation properties because albumin restricted Orange G permeation into NaHA solution of fixed concentration (e.g., 0.2%) to a greater degree than HES of the same concentration as albumin (e.g., 1%).

  1. Experimental proofs for porous NaHA gel structures are necessary. It is hard to believe the porous structures in the conditions suggested in the manuscript.

Response to the comment:

I agree, but unfortunately do not have an experimental setup to prove porous NaHA gel structures in the present study. However, the porous structure of NaHA solutions has been confirmed in previously published studies (see, e.g., review by Fallacara et al., cited as ref. 4).

  1. English editing is necessary.

Response to the comment:

The revised manuscript was edited by a native English speaker.

Reviewer 2 Report

Tatara investigated different effects of albumin and hydroxyethyl starch on low molecular-weight solute permeation through sodium hyaluronic acid solution. The idea and motivation are reasonable, and the characterizations are well performed. However, the following issues must be addressed before it can be published in Polymers.

  1. A schematic summarizing the chemical structure and charge properties of all the materials (HA, albumin, hydroxyethyl starch, and orange G) involving in this study should be added. The interactions between these molecules should be also discussed.
  2. Please provide the PDI of HA used in this study.
  3. In figure 4, why not keep the same concentration of albumin or HES and vary the concentration of HA?
  4. As for HES, at concentrations of 0.1% and 0.2% HA, no restriction of Orange G permeation into NaHA solution was observed. This might be attribute to the difference structure of albumin (spherical) and HES (linear polymer).
  5. Format issue. For example: ‘MW and Mw‘ (line 75-76). ‘rf and φ f’ (line 270)

Author Response

To Reviewer #2:

Thank you for reviewing the manuscript and providing helpful suggestions, which have helped improve the revised manuscript. As described in the point-by-point responses below, all issues raised have been addressed in the revised manuscript.

Tatara investigated different effects of albumin and hydroxyethyl starch on low molecular-weight solute permeation through sodium hyaluronic acid solution. The idea and motivation are reasonable, and the characterizations are well performed. However, the following issues must be addressed before it can be published in Polymers.

  1. A schematic summarizing the chemical structure and charge properties of all the materials (HA, albumin, hydroxyethyl starch, and orange G) involving in this study should be added.

Response to the comment:

As suggested, the schemes of chemical structures of albumin, hyaluronic acid (HA), hydroxyethyl starch (HES), and Orange G have been added as a Supplemental Figure (Figure S1).

  1. The interactions between these molecules should be also discussed.

Response to the comment:

As suggested, brief comments on possible interactions between chemicals have been added to the Discussion section (page 10: line 283-289), as follows:

“One possible explanation for this discrepancy is the difference between the global structure of albumin and HES. First, the spherical structure of albumin may retard Orange G diffusion compared to the linear polymer structure of HES by increasing frictional resistance between Orange G and albumin. Second, the electrostatic interaction of positively-charged amino acid residues of albumin with negatively-charged Orange G could slow Orange G diffusion compared to electrically neutral HES (Figure S1)”

  1. Please provide the PDI of HA used in this study.

Response to the comment:

There is no available data on the polydispersity index (PDI) of HA used in this study. The average molecular weight of NaHA (1.3×106) was obtained from a previous study which used the same material as in the present study (cited as ref. 14) and was calculated from the intrinsic viscosity of NaHA.

  1. In figure 4, why not keep the same concentration of albumin or HES and vary the concentration of HA?

Response to the comment:

I agree that HA concentrations should be altered while fixing the albumin or HES concentration. Nevertheless, in the experimental setup for Orange G permeation into NaHA solutions in the present study, the increase of specific gravity of the medium slows Orange G permeation into NaHA solutions. For example, it was found that Orange G permeation into PBS was slowed compared to permeation into distilled water. On this basis, the total concentration of solutes (i.e., NaHA plus colloids) was forced to be constant so as to minimize the effects of specific gravity of the solution on solute permeation. This is described in the Methods section (page 3: line 88-89). I also believe this issue does not significantly impact the conclusion relating to the different effects of albumin and HES on dye permeation properties because albumin restricted Orange G permeation into NaHA solution of fixed concentration (e.g., 0.2%) to a greater degree than HES of the same concentration as albumin (e.g., 1%).

  1. As for HES, at concentrations of 0.1% and 0.2% HA, no restriction of Orange G permeation into NaHA solution was observed. This might be attribute to the difference structure of albumin (spherical) and HES (linear polymer).

Response to the comment:

Thank you for the helpful comment regarding the potential mechanism underlying the differences in Orange G permeation into NaHA solution between albumin and HES. This potential mechanism was added to the Discussion section (page 10: line 283-286), as follows:

“One possible explanation for this discrepancy is the difference between the global structure of albumin and HES. First, the spherical structure of albumin may retard Orange G diffusion compared to the linear polymer structure of HES by increasing frictional resistance between Orange G and albumin.”

  1. Format issue. For example: ‘MW and Mw‘ (line 75-76). ‘rf and φ f’ (line 270)

Response to the comment:

The manuscript has been thoroughly checked for formatting issues.

Round 2

Reviewer 1 Report

The revision is not satisfactory. Many replies are based on literature, not on experimental results.